

**Technical note: Snow Water Equivalence Estimation (SWEE) Algorithm from**
**Snow Depth Time Series Using a Snow Density Model**
Noriaki Ohara[1*], Siwei He[1], Andrew D. Parsekian[2], and Thijs Kelleners[3]
[1]Department of Civil and Architectural Engineering, University of Wyoming, 1000 E. University Ave.
Laramie, WY 82071
[2]Department of Geology and Geophysics, University of Wyoming, 1000 E. University Ave.
Laramie, WY 82071
[3]Ecosystem Science and Management, University of Wyoming, 1000 E. University Ave.
Laramie, WY 82071
*Corresponding author: nohara1@uwyo.edu

## 12  Abstract

Snow water equivalence (SWE) is typically computed from snow weight by the SNOTEL system in the
US. However, a snow pillow, the main snow weight sensor used by SNOTEL, requires a large, open, flat
area (at least 9 square meters) and substantial maintenance costs.  This article presents the snow water
equivalence estimation (SWEE) algorithm that estimates the SWE evolution merely from continuous
snow depth and temperature measurements using common sensors.  The key component is a depth-
averaged snow density model that is available in the literature, but is underutilized.  Here, we demonstrate
that the snow density model can estimate mass exchanges (SWE changes due to snowfall, erosion,
deposition, and snowmelt) as well as the SWE.  The SWEE algorithm can potentially increase the number
of snow monitoring locations because snow depth and temperature sensors are considerably more
accessible and economical than snow weighing sensor.

## 23  1. Introduction

Snow Water Equivalence (SWE) is important for seasonal forecasting of water resource because it is a
direct indicator of snow water storage. The basic SWE calculation is snow depth multiplied by snow
density. However, in general, it is not straightforward to estimate the season-long evolution of SWE
because snow density is difficult to measure directly and without disturbing the snowpack. The snow
pillow is a common method to weigh snowpack for SWE computation. This sensor requires a relatively
large, open, flat area (3m x 3m, at least), labor-intensive installation, and high maintenance cost. SWE can
also be estimated from the difference between passive, natural electromagnetic energy under and above
snowpack (e.g. Campbell Scientific, CS725). In contrast, a snow depth measurement is easy to measure
continuously using a ranging sensor, even in remote areas since the ranging sensor requires very low
power consumption and minimal maintenance.  If a simple method to determine snow density is available,
SWE could be estimated from the snow depth data without expensive snow pillow or electromagnetic
sensors.
Generally, there are two types of approaches for estimating snow density variation. One uses fitted
equations, typically a function of time (e.g. Kelly et al., 2003; Jonas et al., 2009; Sturm et al., 2010;
Bormann et al., 2014), and the other employs process-based differential equations (e.g. Anderson, 1976).



Kelly et al. (2003) fitted the snow density evolution by a simple logistic curve for processing remote
sensing data. Jonas et al. (2009) and Sturm et al. (2010) modeled snow density by the multiple linear
regressions and the exponential fitting to snow depth, respectively. Bormann et al. (2014) simply used a
linear relationship between mean snow density and the day of year. On the other hand, the snowpack
density and its densification rate may be modeled using more detailed empirical but process-based
relationships (Kojima, 1967, Mellor, 1975, Anderson, 1976, and Lehning et al., 2002). De Michele et al.
(2013) proposed one equation for calculating the bulk snow density, which includes the effects of
compaction, metamorphisms, and snow events on snow density. Among others, one of the available
models was developed by Kuchment et al. (1983), Motovilov (1986), Anderson (1976), Horne and
Kavvas (1997), and completed by Ohara and Kavvas (2006). Although these process-based snow density
models are effective under the changing climate, they have not been used for SWE estimation to date.
This article proposes a spreadsheet-friendly algorithm to extract more useful information from hourly
snow depth and temperature record using the process-based snow density model.
We selected the depth-integrated equation based on the series of the experimental studies on snow
metamorphism processes (Bader et al., 1953; Yoshida, 1955; Kojima, 1967; Mellor, 1974,1975, and
1977), summarized in the supplemental document of this article. The experimental snow density
evolution equation was integrated over a snowpack assuming that the snow density at 2/3 of snow
thickness from the snow surface can represent the average snow density (von der Heydt, 1992).
Accordingly, the depth-averaged snow density evolution may be described by the following ordinary
differential equation,
$$\frac{d\rho_s}{dt} = \frac{2}{3\eta_o} D\rho_s e^{-0.04(T_c-T_s)} e^{-k_o\rho_s} - \frac{(\rho_s-\rho_{ns})\rho_w}{D\rho_{ns}} sn \ .$$   (1)

$t$ = time (hour)
$D$ = snow depth (cm)
$\rho_s$ = snowpack density (g/cm$^3$)
$\rho_{ns}$ = new snow density (g/cm$^3$)
$\rho_w$ = density of liquid water = 1.0 (g/cm$^3$)
$T_c$ = critical temperature = 0°C
$T_s$ = snow surface temperature (°C)
$k_0$ = coefficient = 21.0 (cm$^3$/g) (Kojima 1967, Anderson et al. 1976)
$\eta_0$ = viscosity coefficient, a constant at 0 °C = 15.0 ~ 38.0 (cm·hour) (Kojima 1967,
Anderson et al. 1976)
$sn$ = rate of incoming snow (cm/hour)

The exponential term in the right hand side describes the gravitational compaction-snow metamorphism,
and the second term expresses the density change created by the new snow deposit. Note that the unit
system was restricted in the gram-centimeter-hour system by the original empirical relationships in the
literature. Here, we adopt this convenient depth-integrated equation because it sufficiently covers the
major snow densification processes as well as being simple.

## 2. Method

First, the snow density equation, Equation (1), can be rewritten as

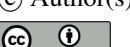



$$\frac{d\rho_s}{dt} = G(D, \rho_s, T) + F(D, \rho_s, sn),  \quad (2)$$

where
$$G(D, \rho_s, T) = \frac{2}{3\eta_o} D\rho_s e^{0.04T_s} e^{-k_o\rho_s},  \quad (3)$$

$$F(D, \rho_s, sn) = -\frac{(\rho_s - \rho_{ns})\rho_w}{D\rho_{ns}} sn.  \quad (4)$$

The SWE change per unit time, $\Delta SWE$ (cm/hour), can be estimated as the difference of Snow Water
Equivalences (SWEs) at future time and present time. That is,
$$\Delta SWE = \widehat{SWE}^{i+1} - SWE^i,  \quad (5)$$

where the superscript denotes time step, and variable with hat is predicted quantity without SWE
deposition or abrasion. The SWE at the present time i can be computed as,
$$SWE^i = D^i \frac{\rho_s{}^i}{\rho_w}.  \quad (6)$$

Similarly, the SWE at the one time step ahead i+1 can be computed as,
$$\widehat{SWE}^{i+1} = D^{i+1} \frac{\widehat{\rho_s}^{i+1}}{\rho_w},  \quad (7)$$

where the predicted snow density due only to gravitational compaction may be obtained from Equation
(3)

$$\widehat{\rho_s}^{i+1} = \rho_s{}^i + \Delta t \cdot G(D^i, \rho_s{}^i, T_s{}^i).  \quad (8)$$

When the change $\Delta SWE$ is negative, the snowpack loses its mass by snowmelt, wind erosion, or
sublimation. If the change $\Delta SWE$ is positive, the snowpack received snowfall or snowdrift. In this case,
the depth integrated snow density must be adjusted with new snow density.
$$\begin{cases} \rho_s{}^{i+1} = \widehat{\rho_s}^{i+1} + \Delta t \cdot F(D^i, \rho_s{}^i, \Delta SWE) & ; \Delta SWE > 0 \\ \rho_s{}^{i+1} = \widehat{\rho_s}^{i+1} & ; \Delta SWE \leq 0 \end{cases}.  \quad (9)$$

Note that this algorithm requires two assumptions: 1) no snowmelt occurs during snow fall; 2) the density
of new snow and drifted snow are the same.
Additionally, snow depth data must be preprocessed by following two steps: 1) removing the
measurement oriented abrupt jumps and 2) removing the random noise in the ranging sensor. The latter
data process can be performed by either moving average or exponential smoothing filter. In this study,
exponential smoothing filter was selected because filtering factor can be easily adjusted even in a
spreadsheet. The exponential smoothing filter may be expressed as,
$$D_f^i = \alpha D^i + (1 - \alpha)D_f^{i-1},  \quad (10)$$



where $D_f^i$ is the filtered snow depth data at the present time i, and $\alpha$ is the smoothing factor between 0 and
1. The smoothing factor may be selected depending on the data error or noise level, and will be discussed
in the following section.

## 3. Results and Discussion

We applied the SWEE algorithm to data collected at the Brooklyn Lake SNOTEL site (41.3666 N,
106.2333W, 3122 m ASL) in the Snowy Range, Medicine Bow National Forest, Wyoming. Figure 1
shows the time series of the observed snow depth (upper panel), the model-estimated snow density
(second panel), the observed and computed SWE, and the computed SWE fluxes. The snow density
model described above was calibrated using three parameters ($\rho_{ns}, \eta_o$, and $\alpha$) for the snow density data
by the snow pit survey, denoted in the dot in the second panel of Figure 1, and for the SWE estimates of
the SNOTEL. The parameters selected for this demonstrative application were: $\rho_{ns} = 0.18$ (g/cm³), $\eta_o =$
21 (cm·hour), $\alpha = 0.1$. The snow temperature was approximated by the hourly air temperature record in
this application. The SNOTEL SWE data, denoted by SWE obs., which were computed from the snow
depth and snow pillow data by the Natural Resources Conservation Service (NRCS) with the standard
procedure, was considered to be "ground truth" in this article.
The SWEE algorithm output, denoted by SWE est., was comparable to the SNOTEL SWE, denoted by
SWE obs., from the third panel of Figure 1 (Nash–Sutcliffe model efficiency [NSME] = 0.864 and R² =
0.854) (Nash and Sutcliffe, 1970). We find that the timing and magnitude of noticeable changes in SWE
is well represented. However, there was some discrepancy in the SWE comparison, including over-
prediction of SWE before 18 March 2015 and under-prediction of SWE after 18 March 2015. We pose
seven possible reasons for these departures from measured values:
1. Snow density of newly fallen snow $\rho_{ns}$ is time dependent
2. Snow erosion and abrasion by wind affect the snow depth observation
3. Snow pillow measurement brings error into the SWE obs.
4. Air temperature substitution for snow temperature is questionable
5. The depth-averaged snow density model is overly simplified
6. Accuracy of the snow depth measurement is poor
7. Treatment for the snow depth data is inappropriate
There could be different reasons that lead to the discrepancy; however, we found that the most important
one may be the new snow density effect (item no.1). Figure 2 shows the sensitivity analysis result for
three main model parameters. The NSME coefficient for the SWE was used for the model performance
evaluation here. Clearly, new snow density $\rho_{ns}$ is the most sensitive parameter among three parameters
($\rho_{ns}, \eta_o, \alpha$). It makes sense that new snow density varies from early winter storms to spring snow events
(e.g. Susong et al., 1999). This means that, if accuracies of the all observed data are ensured, it might be
possible to analyze the density variation of newly fallen snow using this algorithm in theory. The effect of
the air temperature substation for snow temperature (item no.4) could not be evaluated due to lack of the
suitable sensor. On the other hand, the SWE estimation was found to be surprisingly insensitive to the
exponential smoothing factor $\alpha$.





Figure 1 also includes the selected regression model estimates using a power function fitting (Equation
(1) in Jonas et al., 2009), multiple linear fittings (Equation (4) in Jonas et al., 2009), and a time adjusted
exponential function fitting (Equation (6) in Sturm et al., 2010). The computed statistics, $R^2$ and NSME,
to on the SWE are tabulated in Table 1. It was found that the regression models can capture the mean of
the SWE dynamics well while the rapid changes of the snow density after the snow accumulation events
could not be expressed. The SWEE algorithm was able to reproduce the overall shape of the SWE time
series although the SWE estimation by the SWEE was less accurate than those by the regression models.
This is because the SWEE algorithm relies on the predictive snow density change model; therefore, the
prediction error of each time step accumulates in the SWE estimate. However, this could lead to the
better SWE change estimation (snow accumulation and abrasion) than the regression models during the
snow storms because the snow density estimates of the regression models tend to represent the longer
period average. More importantly, the SWEE algorithm is based on the process-based snow density
model, which is independent from climate, weather, and region.
To further test the performance of the SWEE algorithm, was applied at the Noname (NN) Creek research
site (41.3437 N, 106.2121W, 2950 m ASL). The model was recalibrated using a snow density
measurement by a snowpit survey on 9 April 2018 near the site, and the parameters were determined as:
$\rho_{ns} = 0.22$ (g/cm³), and $\eta_o = 21$ (cm·hour). The site was instrumented with a snow lysimeter, ranging
snow depth sensor, and snow surface temperature, but there is no snow pillow at the NN Creek research
site. More detailed information regarding this site is available in Pleasants et al. (2017), Thayer et al.
(2018), and He et al. (2018). Figure 3 shows the computed SWE change $\Delta SWE$ during the snowmelt
season of 2018, which includes the lysimetric snowmelt water flux at the bottom of snowpack, denoted in
the blue-shaded area. The effect of the smoothing factor $\alpha$ on the SWEE method output can be visualized.
When the snow depth data is under-smoothed with $\alpha=0.8$, unrealistic snow gains during the night
appeared throughout the snowmelt period, probably due to measurement error. This suggests that the
snow ranging sensor has been influenced by local atmospheric conditions affecting ultrasonic wave
propagation. On the other hand, the over-smoothed case with $\alpha=0.1$ resulted in continuous snowmelt
even during the night time, which is unrealistic, because the lysimetric snow water flux had higher peaks
than the SWEE snowmelt rate. Clearly, SWE change estimates, a byproduct of the SWEE algorithm, are
highly dependent on the smoothing factor, $\alpha$, although the SWE estimation was not very sensitive to $\alpha$.
Thus, the smoothing filter is required for snow depth data by a conventional sonic ranging sensor (such as,
SR50A-L, Campbell Scientific) in order to eliminate false SWE gains, especially for computing daily-
total snowmelt rate.
Additionally, the phase of the peaks can be analyzed using this SWEE change estimate. It is rational to
have larger phase difference between the snow lysimeter data and snowmelt (SWE change) in the early
snowmelt season because of the thicker snowpack. However, the phase differences were found to be
irregular from day to day suggesting that the snow percolation process extremely complicated due to
thermal and water flow effects on the snow matrix.

## 4. Conclusions

This article described the SWEE algorithm that uses the relatively reliable time series data of snow depth
and temperature. This method does not rely on snow energy flux computations or empirical snowmelt
equation (e.g. temperature index equation), which required significantly more measurements, assumptions,





and often model calibrations.  As such, the SWEE algorithm has fewer uncertainties than full snow
modeling. The small data requirements make this algorithm particularly effective for situations were data
is scarce and is a useful tool for quality verification of the existing snow pillow measurements at the
SNOTEL sites. The SWE change, which is byproduct of this algorithm, can potentially quantify wind
snow erosion, deposition, and sublimation at a very local scale after careful snow density model tuning.
To avoid unrealistic SWE exchanges when integrating over a long period, an exponential smoothing filter
is recommended to correct the long term mass balance estimates.  However, the choice of the parameter
of the exponential smoothing filter must depend on the purpose of the analysis. We recommend additional
snow surveys (e.g. snow pit, snow course) that provide model calibration opportunities and some
confidence in the SWE and SWE change estimations.
Lastly, approximation of snow surface temperature by air temperature may not be suitable depending on
the energy flux fraction.  Thus, this study suggests that snow surface temperature be added to any snow
monitoring system because it is more appropriate than air temperature for SWE estimation.

## Acknowledgements

This study was supported by the National Science Foundation EPS-1208909.  The spreadsheets for the
SWEE algorithm computations are available as supplementary document.

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





**Table 1 – Statistics of the model fittings with respect to SWE at the Brooklyn Lake SNOTEL site during the**
**water year 2015**

| Snow density model | $R^2$ | NSME | Note |
|---|---|---|---|
| **SWEE (this study)** | 0.864 | 0.816 | Process-based snow density model |
| **Jonas et al. 2009, Eqn.(1)** | 0.726 | 0.718 | Regression to snow depth with a power function with a single parameter |
| **Jonas et al. 2009, Eqn.(4)** | 0.937 | 0.933 | Regression to snow depth with a linear function with monthly parameters |
| **Sturm et al. 2010, Eqn.(6)** | 0.911 | 0.859 | Regression to snow depth with an exponential function with day-of-year adjustment |




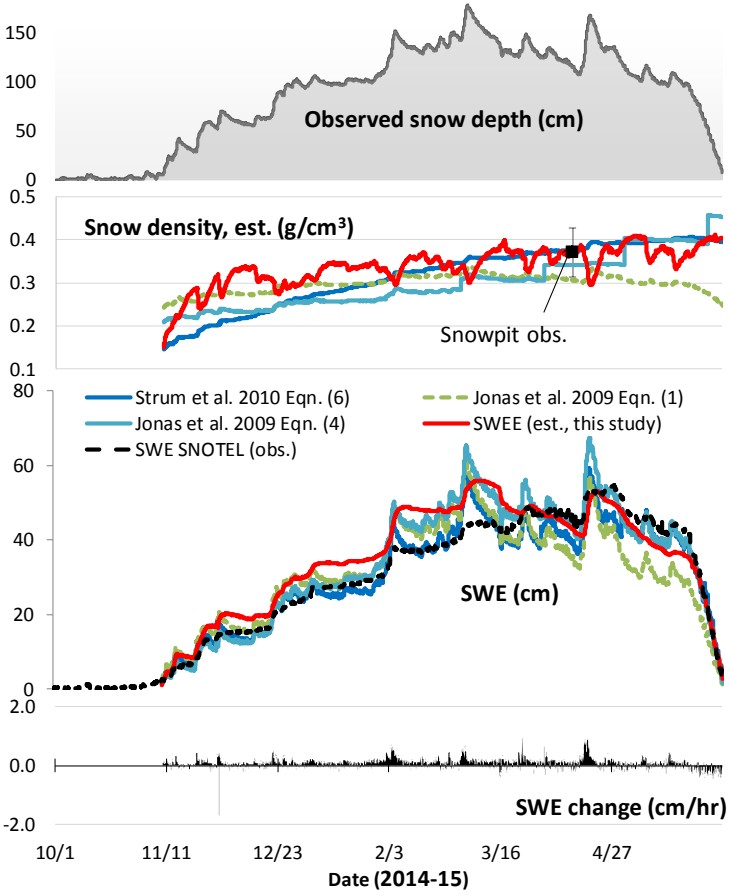

Figure 1 - SWEE application example at the Brooklyn Lake SNOTEL site during the water year 2015. Three regression models (Strum et al., 2010; Jonas et al., 2009) are included in the snow density and SWE graphs in the middle two panels.





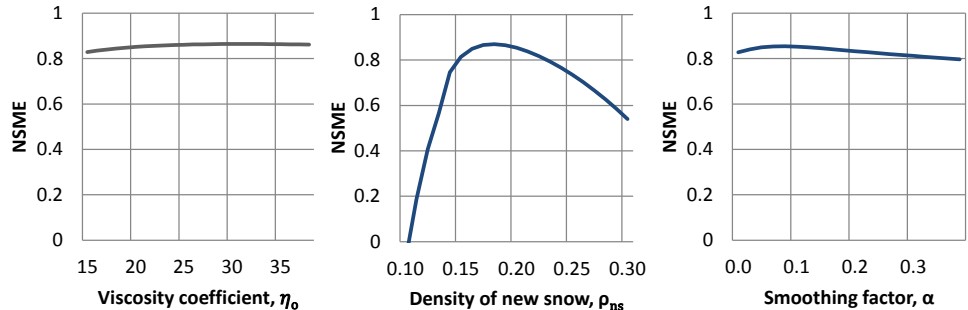

**Figure 2 - Sensitivity analysis of the snow density model for SWE estimation**





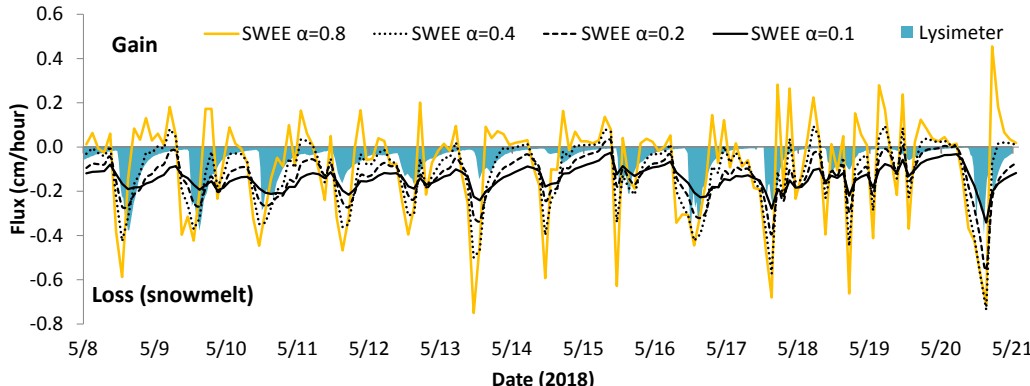

**Figure 3 - SWE change computed by the SWEE algorithm and corresponding the lysimetric snowmelt water flux at the bottom of the snowpack in the NN research site**