# Peer review of "Technical note: Snow Water Equivalence Estimation (SWEE) Algorithm from"

_Hydrology and Earth System Sciences, 2018_

## Referee Comment (RC1) · Anonymous Referee #1 · 29 Oct 2018

This technical note presents a method that brings together previous research on snow density as it relates to depth. The authors compare the estimations to a single year at a SNOTEL station as well as a single year with a snowmelt lysimeter.

Overall, I think the idea has merit and could be a useful tool when a more complex computational model is not appropriate. However, I think that the manuscript in it's current form lacks enough validation or error analysis for the presentation of a new method. The comparison of the method to a single station for a single year is not enough; I would really like to see multiple locations with varying conditions and snow years to be convinced. Furthermore, I think that the comparisons to other models is

lacking. While I am happy to see the comparisons to Jonas et al., 2009 and Sturm et al., 2010, much has been done in the past 8-9 years. Perhaps a comparison to a more complex model like SnowPack in addition to what is already shown. That could provide evidence that if all you need are bulk properties that this new approach could be just as good and computationally easier.

More specific points are as follows:

I appreciate the listing of 7 possible reasons for the errors, but these should really be better quantified in magnitude of error and some can actually be directly addressed with the available data.

For example: 1. snow density of newly fallen snow: The authors show the sensitivity of the equation to this parameter, but that doesn't mean this parameter is the reason for the error. Furthermore, enough information is available to compare what the SWEE estimation is to the Snotel new snow density (depth change during storm and SWE increase can be used to determine new snow density).

2. snow erosion and abrasion affect snow depth: By how much? what is the impact on the estimation (quantify)

3. Snow pillow measurement error - again, how much? There is literature on this, it will depend on the time of the season, etc. but how will this impact your comparison? To this end, more snow pit observations would go quite far in this study.

4. air temperature substitution - how much does this impact your results?

6. Accuracy of snow depth - this can be addressed a couple of ways. First there is literature on this and/or manufacturer defined accuracies that should be taken into account. Second, quantification of the theoretical error from this could easily be done. If the depth is off by 2 cm, how much does that change the SWEE result? Is this different for shallow or deep snowpacks?

7. Treatment for snow depth is inappropriate - If this is the case, first quantify the

effects, then second why not change the treatment? When applying a new method like this I think the data pre-processing can be huge. If your input is bad then your output will be bad. So why do you think the treatment may be inappropriate and how can you change it? Then do this to prove the presented method.

line 153: the authors state that "However, this could lead to better SWE change estimation" talking about the thought that SWEE may be better than the other regression models to estimate melt, then the manuscript compares the new method to lysimeter data, but not the other models. With a claim that the new method may be better, it should really be shown because you have the data to test this statement.

According to table 1, there are two other methods that work better and are simpler to apply, please argue convincingly as to what the best method is and why (with data).

Fig. 1 - labeling each panel would help a lot. Why is the SNOTEL density not included in the density panel? This could help show how much better/worse each method is for estimating density.

Throughout the paper the authors use "change [delta]SWE", but the [delta] symbol is generally used to denote "change in". Is it supposed to be something different here, please double check.

please correct "Strum et al." in figure 1 to "Sturm et al."

the term "SWEE" is confusing to me, especially when the wording went back and forth with SWE observations comparing to SWEE. Perhaps changing it to "estimation of SWE (eSWE) to help the reader.

I hope that these comments are helpful.
* * *

---

## Referee Comment (RC2) · Jonas (Referee) · 4 Dec 2018

This technical note reports on using an existing snow density model to derive SWE from a temporally continuous record of snow depth. Given the effort required to operate and maintain a snowpillow, being able to estimate SWE from alternative snow depth measurements has its potential uses and merits, e.g. for gap filling purposes, or in case other meteorological data being unavailable to run a full snowpack model.

Obviously the authors were not the first to come up with using a snow density model in that particular context. The performance is similar (or in fact slightly worse) in comparison to two parametric models that were developed about 10 years ago with the

same application in mind. However, unlike those two alternative offerings the approach presented here is capable of providing meaningful time series of SWE at high temporal resolution (at the cost of requiring complete time series of snow depth and temperature input data). This would be a good selling point of the paper if it wasn't for other publications that have already tackled this very aspect, see e.g. the excellent paper by McCreight and Small from 2014 (doi.org/10.5194/tc-8-521-2014).

Now the question arises, what then is the selling point of this paper? As a technical note, it might suffice to present the method as an alternative approach – if the authors manage to identify at least some differences to existing models. Better transferability to other sites without recalibration maybe? And obviously, testing the model with data from one season and one SNOTEL station only is not nearly enough.

I suggest the authors put more effort in precisely framing their work in the context of alternative snow density models; and perform a multi-site and multi-season validation.

Below just a few more specific comments:

Line 1: the title is somewhat misleading given that your model also requires temperature data as input

Line 48-51: extend this into a convincing last paragraph of your introduction, which highlights shortcomings of existing models a/o the merits of your approach (to be demonstrated below)

Line 52: start method section here

Line 101: and the former?

Line 117: while this assumption is what you often have to work with, it would be interesting to also deploy your model at a site where you actually do have snow temperature data to test, how big of a problem is this assumption?

Line 127: since you require temperature data as model input anyway, why not using a

new snow density formulation that is temperature dependent?

Line 156: ". . . which is independent from climate ..."? You may have a point there, but let's first see what happens if you apply your model in Japan, Siberia, Lesotho; without recalibration of course. You can refer to Matthew Sturm's snow classification scheme to back up your claim.

---

## Author Comment (AC1) · 20 Dec 2018

Thank you for reviewing our manuscript. The responses to the comments are inserted below in blue color.

Overall, I think the idea has merit and could be a useful tool when a more complex computational model is not appropriate. However, I think that the manuscript in it's current form lacks enough validation or error analysis for the presentation of a new method. The comparison of the method to a single station for a single year is not enough; I would really like to see multiple locations with varying conditions and snow years to be convinced. Furthermore, I think that the comparisons to other models is

lacking. While I am happy to see the comparisons to Jonas et al., 2009 and Sturm et al., 2010, much has been done in the past 8-9 years. Perhaps a comparison to a more complex model like SnowPack in addition to what is already shown. That could provide evidence that if all you need are bulk properties that this new approach could be just as good and computationally easier.

More complex snow model requires energy flux estimations, which typically functions of air temperature, humidity, radiation, wind speed, precipitation, soil temperature, etc., while this technique (SWEE algorithm) just estimates the SWE from the snow depth and temperature data. They are not comparable in our opinion. In fact, snow depth is one of the output variables of the complete snow model such as SnowPack model, which should be used when high-quality atmospheric forcing data are available. However, this simple technique is effective when the atmospheric data is insufficiently available in the historical sites.

We do not think that additional application is required for this **technical note** article. However, we applied it to four selected SNOTEL sites for 2016, 2017, and 2018 water years (total 12 years). The sites were arbitrarily chosen from the US SNOTEL with snow pillow in four regions (Pacific Northwest, Central Sierra Nevada, Northeast, and Alaska). The results are shown in the end of this reply document. Note that that all parameters except new snow density were kept the same for these computations. The results indicate that the SWEE algorithm can sometimes get better than the other data-driven approaches depending on the data quality.

I appreciate the listing of 7 possible reasons for the errors, but these should really be better quantified in magnitude of error and some can actually be directly addressed with the available data.

For example: 1. snow density of newly fallen snow: The authors show the sensitivity of the equation to this parameter, but that doesn't mean this parameter is the reason for the error. Furthermore, enough information is available to compare what the SWEE estimation is to the Snotel new snow density (depth change during storm and SWE increase can be used to determine new snow density).

We intend to show that the new snow density was very sensitive to the SWE estimation. Please note that the change of snow depth is the main part of the SWEE algorithm.

2. snow erosion and abrasion affect snow depth: By how much? what is the impact on the estimation (quantify)

This is an open question. Combining this methodology and some accurate snowmelt estimation may provide an opportunity to separate erosion and melt.

3. Snow pillow measurement error - again, how much? There is literature on this, it will depend on the time of the season, etc. but how will this impact your comparison? To this end, more snow pit observations would go quite far in this study.

This is beyond the scope of this Technical Note article, which should be concise.

4. air temperature substitution - how much does this impact your results?

Unfortunately, we do not have air temperature and snow surface temperature, concurrently. So, we could not quantify it.

6. Accuracy of snow depth - this can be addressed a couple of ways. First there is literature on this and/or manufacturer defined accuracies that should be taken into account. Second, quantification of the theoretical error from this could easily be done. If the depth is off by 2 cm, how much does that change the SWEE result? Is this different for shallow or deep snowpacks?

The SWEE algorithm relies on the snow depth difference from the one time step prior. So, the fluctuation in the snow depth data would affect the SWEE estimation.

Assuming snow density is 0.30 g/cm3, 2 cm error in snow depth should result in 0.6 cm error in SWE.

7. Treatment for snow depth is inappropriate - If this is the case, first quantify the effects, then second why not change the treatment? When applying a new method like this I think the data pre-processing can be huge. If your input is bad then your output will be bad. So why do you think the treatment may be inappropriate and how can you change it? Then do this to prove the presented method.

No matter what methodology you choose, the data cleaning/preprocessing is necessary. Otherwise, we end up with the "garbage in, garbage out" situation. To illustrate the case without exponential smoothing filter ( $\alpha = 1.0$ ), we adjusted the Figure 3 (see below). We think that the treatment for snow depth using the filter is appropriate and effective.

line 153: the authors state that "However, this could lead to better SWE change estimation" talking about the thought that SWEE may be better than the other regression models to estimate melt, then the manuscript compares the new method to lysimeter data, but not the other models. With a claim that the new method may be better, it should really be shown because you have the data to test this statement.

We computed the other methods in the NN. The results are shown below. We accordingly revised the text since the other models are as good as the SWEE algorithm.

The same graph with longer period is shown below. Due to the very large unrealistic variations in the snow depth measurement during the snow accumulation period, none of the models can accurately estimate the SWE change from this dataset except the snowmelt period.

---

## Author Comment (AC2) · 20 Dec 2018

Thank you, Dr. Jonas, for your valuable comments on our work. The responses to the comments are inserted below in blue color.

This technical note reports on using an existing snow density model to derive SWE from a temporally continuous record of snow depth. Given the effort required to operate and maintain a snowpillow, being able to estimate SWE from alternative snow depth measurements has its potential uses and merits, e.g. for gap filling purposes, or in case other meteorological data being unavailable to run a full snowpack model.

Obviously the authors were not the first to come up with using a snow density model in that particular context. The performance is similar (or in fact slightly worse) in comparison to two parametric models that were developed about 10 years ago with the same application in mind. However, unlike those two alternative offerings the approach presented here is capable of providing meaningful time series of SWE at high temporal resolution (at the cost of requiring complete time series of snow depth and temperature input data). This would be a good selling point of the paper if it wasn't for other publications that have already tackled this very aspect, see e.g. the excellent paper by McCreight and Small from 2014 (doi.org/10.5194/tc-8-521-2014).

We included the McCreight and Small (2014) in the review section. The existing snow density models referred in the comments (Kelly et al., 2003; Jonas et al., 2009; Sturm et al., 2010; Bormann et al., 2014; McCreight and Small, 2014) are all data-driven approach. Meanwhile, the model proposed in this article clearly belongs to a process-based approach as described in the supplement. I paste the derivation of the model for your convenience. This model is accumulation of experimental works by snow scientists, which have been largely ignored or not utilizes (to my view, N.Ohara).

**Supplement of "Technical note: Snow Water Equivalence Estimation (SWEE) Algorithm from Snow Depth Time Series Using a Snow Density Model"**

Noriaki Ohara
Civil and Architectural Engineering, University of Wyoming
Phone:+1(307)343-2670; nohara1@uwyo.edu

**Derivation of snow compaction term**

Yoshida (1955) formulated the snow density change due to gravity and snow metamorphism effect.

$$\frac{1}{\rho}\frac{d\rho}{dt} = \frac{w_s}{\eta} \qquad (S\text{-}1)$$

$w_s$=weight of snow above the layer in terms of SWE (cm)
$\eta$=viscosity coefficient, a constant for a given density and temperature (cm·hr)
$\rho(z, t)$ = snow density at depth z at time t (g/cm$^3$)

The viscosity coefficient includes two different effects.

$$\eta = \eta_c \eta_t \qquad \text{(S-2)}$$

$\eta_c$=viscosity coefficient for gravitational compaction
$\eta_t$=viscosity coefficient for temperature change

Kojima (1967) obtained following expression from experiments:

$$\eta_c = \eta_{co} \exp[k_o \rho] \qquad \text{(S-3)}$$

Mellor (1975) formulated viscosity coefficient for temperature change as follows.

$$\eta_t = \eta_{to} \exp\left[\frac{A}{R} \frac{(T_c - T)}{T T_c}\right] \qquad \text{(S-4)}$$

Anderson (1976) simplified it as,

$$\eta_t = \eta_{to} \exp[0.08(T_c - T)] \qquad \text{(S-5)}$$

Combining (S-1) through (S-5) yields,

$$\frac{1}{\rho}\frac{d\rho}{dt} = \frac{w_s}{\eta_o} \exp[-0.08(T_c - T)] \exp[-k_o \rho] \qquad \text{(S-6)}$$

where

$$\eta_o = \eta_{co} \eta_{to} \qquad \text{(S-7)}$$

Assuming the depth averaged snow density $\rho_s$ corresponds at the 2/3 of the snow depth from the surface (von der Heydt, 1992),

$$w_s = \frac{2}{3} D \qquad \text{(S-8)}$$

The predictive ordinary differential equation form of the depth averaged snow density can be obtained by the depth integration (Horne and Kavvas, 1997)

$$\frac{d\rho_s}{dt} = \frac{2 D \rho_s}{3 \eta_o} \exp[-0.08(T_c - \bar{T})] \exp[-k_o \rho_s] \qquad \text{(S-9)}$$

$\rho_s$ = depth averaged snow density
$\bar{T}$ = depth averaged snow density

Assuming a triangular temperature profile in a snowpack, which indicates snow surface temperature $T_s = 2\bar{T}$, the compaction term in Equation (1) can be obtained.

**Derivation of new snow term**

When new snow sits on the snow surface, snow density must be modified as below:

$$\rho_s + \Delta\rho_s = \frac{D\rho_s + \Delta D \rho_{sn}}{\Delta D + D} \qquad \text{(S-10)}$$

$\rho_s$ = depth averaged snow density (g/cm$^3$)
$\Delta\rho_s$= snow density change (g/cm$^3$)
$D$= snow depth (cm)
$\Delta D$= snow depth change (cm)
$\rho_{sn}$ = new snow density (g/cm$^3$)

$$\rho_s + \Delta\rho_s = \frac{D\rho_s + \Delta D\rho_s - \Delta D\rho_s + \Delta D\rho_{sn}}{\Delta D + D}$$

$$\rho_s + \Delta\rho_s = \frac{D\rho_s + \Delta D\rho_s}{\Delta D + D} + \frac{-\Delta D\rho_s + \Delta D\rho_{sn}}{\Delta D + D}$$

$$\rho_s + \Delta\rho_s = \rho_s + \Delta D\frac{(\rho_{sn}-\rho_s)}{\Delta D + D}$$

$$\Delta\rho_s = \Delta D\frac{(\rho_{sn}-\rho_s)}{\Delta D + D} \tag{S-11}$$

Now, snow depth change can be expressed in terms of snowfall rate $sn$ (cm/hr) as follows.

$$\Delta D = \Delta t \cdot sn \cdot \frac{\rho_w}{\rho_{sn}} \tag{S-12}$$

Substituting Equation (S-12) to Equation (S-11) yields

$$\Delta\rho_s = \Delta t \cdot sn \cdot \frac{\rho_w}{\rho_{sn}}\frac{(\rho_{sn}-\rho_s)}{(\Delta D + D)} \tag{S-13}$$

$$\frac{\Delta\rho_s}{\Delta t} = \frac{\rho_w}{\rho_{sn}}\frac{(\rho_{sn}-\rho_s)}{(\Delta D + D)}sn \tag{S-14}$$

When $\Delta t$ is sufficiently small, and $\Delta D \ll D$, one can obtain,

$$\frac{d\rho_s}{dt} = -\frac{(\rho_s-\rho_{sn})\rho_w}{\rho_{sn}D}sn \tag{S-15}$$

Now the question arises, what then is the selling point of this paper? As a technical note, it might suffice to present the method as an alternative approach – if the authors manage to identify at least some differences to existing models. Better transferability to other sites without recalibration maybe? And obviously, testing the model with data from one season and one SNOTEL station only is not nearly enough.

Please refer to the previous response. This is essentially a different type of model from the "existing" models.

Line 1: the title is somewhat misleading given that your model also requires temperature data as input

We revise the title:

Technical note: Snow Water Equivalence Estimation (SWEE) Algorithm from Snow Depth and Temperature Time Series Using a Snow Density Model

Line 48-51: extend this into a convincing last paragraph of your introduction, which highlights shortcomings of existing models a/o the merits of your approach (to be demonstrated below)

Line 52: start method section here

Line 101: and the former?

We re-organized the sections accordingly. Thank you. That is a really good suggestion.

Line 117: while this assumption is what you often have to work with, it would be interesting to also deploy your model at a site where you actually do have snow temperature data to test, how big of a problem is this assumption?

Unfortunately, we do not have concurrent snow and air temperature data. Also, this is beyond the scope of this small article.

Line 127: since you require temperature data as model input anyway, why not using a new snow density formulation that is temperature dependent?

It is a good idea. We adjusted the snow density as a linear function of the temperature data.

$$\rho_{ns} = \rho_{nso} + aT_{obs}$$

When $\rho_{nso}$=0.167 (intersect) and $a$=0.0034 (slope), we obtained better comparison as shown below. This analysis indicates that the depth averaged snow density model can describe the snow density evolution if incoming snow density is correctly estimated. In this case, the SWEE algorithm performed better than the other data-driven models, as shown below. This implies that the main source of our estimates is the density of new snow. However, we do not intend to include it in the main article in order to keep this technical note article simple.

[Figure]

| Snow density model | $R^2$ | NSME | Note |
|---|---|---|---|
| SWEE (this study) | 0.927 | 0.918 | Dynamic snow density model |
| Jonas et al. 2009, Eqn.(1) | 0.716 | 0.707 | Regression with a power function |
| Jonas et al. 2009, Eqn.(4) | 0.931 | 0.927 | Regression with a linear function with monthly p |
| Sturm et al. 2010, Eqn.(6) | 0.906 | 0.852 | Regression with an exponential with day-of-year |

Line 156: "… which is independent from climate …"? You may have a point there, but let's first see what happens if you apply your model in Japan, Siberia, Lesotho; without recalibration of course. You can refer to Matthew Sturm's snow classification scheme to back up your claim.

The process-based approach presented here is inherently such an attribute unlike the data-driven approach. However, the purpose of this article is not proving it. We added some additional model applications in Washington, California, Vermont, and Alaska in the response letter for the RC1. It shows that the SWEE

algorithm works okay and often great for these sites as well.   We unfortunately could not secure sufficient time/data for the regions you suggested.

---

## Author Comment (AC3) · 20 Dec 2018

The supplement file encloses the revised manuscript.

Please also note the supplement to this comment:
https://www.hydrol-earth-syst-sci-discuss.net/hess-2018-451/hess-2018-451-AC3-supplement.zip

---

## Editor Comment (EC1) · Seibert (Editor) · 23 Dec 2018

Thanks for submitting this manuscript to HESS. We have received two excellent reviews. Both appreciate the work, but also raise important concerns and in the end recommend major revisions respective rejection. Based on the comments, and my own reading, the novelty of the presented work is not fully clear. The argument that the presented model might be more robust for applications in different regions is interesting and possibly reasonable, but this needs to be demonstrated. One might argue that the presented approach is more process based than some of the recent work on SWE estimation or snow density modelling (Kelly et al., 2003; Jonas et al., 2009; Sturm et

al., 2010; Bormann et al., 2014; McCreight and Small, 2014), but the difference seems rather gradual. The presented approach is based on several assumptions that help to reduce the need for input data. While the latter has its clear value, this also means that the approach, in the end, might not represent the processes correctly after all. Therefore the question of the novel aspects of the presented work as raised by both reviewers is a very valid concern. The authors were apparently not aware of the paper by McCreight and Small (2014). This is unfortunate, but given the volume of publications, such things happen. However, I do not think that just adding a reference is making justice to this important comment by the reviewer. What is asked for is to relate the presented work to the model being presented by McCreight and Small (2014)!

An test using several sites as suggested by one of the reviewers and at least partly been presented in the authors' response would be a valuable addition to this work and make a good argument for the new model. It also seems that the structure is suboptimal with important information being hidden in the supplementary material.

Overall, I find the authors' responses a bit on the defensive side and would like them to consider the valuable reviewer comments in a constructive way in their revisions of the manuscript to make the best use out of these comments for improving this manuscript.

As a small note, the open review process in HESS allows posting responses directly when reviewer reports come in. In this case, an earlier response would have been good to allow for some discussion. Please also note, that the revised version should not be submitted as the authors' response, but after the discussion phase has ended and the editor has looked at the reviews and responses.

Best regards, Jan Seibert